# Conversion of Hemiarthroplasty to Reverse Shoulder Arthroplasty with Humeral Stem Retention

**DOI:** 10.3390/jcm11030834

**Published:** 2022-02-04

**Authors:** Falk Reuther, Ulrich Irlenbusch, Max J. Kääb, Georges Kohut

**Affiliations:** 1Department of Trauma Surgery and Orthopaedics, DRK Kliniken Berlin Koepenick, Salvador-Allende-Str. 2-8, 12559 Berlin, Germany; 2Clinic for Orthopaedics and Trauma Surgery, Erfurt Sports Clinic, Am Urbicher Kreuz 7, 99099 Erfurt, Germany; prof.irlenbusch@sportklinik-erfurt.de; 3Clinic for Sports Medicine and Orthopaedics, Sporthopaedicum Straubing, Bahnhofplatz 27, 94315 Straubing, Germany; max.kaeaeb@gmx.de; 4Clinic for Orthopaedics and Trauma Surgery, Clinique Générale Fribourg, Rue Hans Geiler 6, 1700 Fribourg, Switzerland; gkohut@cliniquegenerale.ch

**Keywords:** reverse shoulder arthroplasty, conversion, failed hemiarthroplasty, shoulder hemiprosthesis, modular reverse prosthesis

## Abstract

The purpose of this study is to evaluate the mid-term clinical results of an ongoing case series on conversion reverse shoulder arthroplasty (RSA) with a modular prosthesis system. We included 17 elderly patients revised for failed hemiarthroplasty after proximal humeral fracture, of which 13 were converted using a modular reverse shoulder prosthesis. Four could not be converted due to overstuffing. For the conversion RSA, we determined the Constant score, American Shoulder and Elbow Surgeons Shoulder Score, visual analogue scale for pain and satisfaction, and range of motion preoperatively, at one year, and at the last follow-up. All measured clinical outcomes improved significantly at both follow-up time points (*p* < 0.05). The mean duration of surgery was 118.4 min (range: 80.0 to 140.0 min). We observed complications in three patients; these included one late infection and two aseptic stem loosenings. Modular shoulder arthroplasty is a suitable procedure for conversion RSA in elderly patients. All measured postoperative clinical outcomes improved significantly, the complication rate was acceptable, and no prosthesis-related complications occurred. Conversion RSA, although not feasible in every case, is a viable treatment option in the elderly, which can provide successful mid-term results.

## 1. Introduction

Proximal humeral fractures in elderly patients are a common indication for shoulder arthroplasty. Treatment options mainly involve hemiarthroplasty (HA) and reverse shoulder arthroplasty (RSA). Choosing the appropriate treatment option is influenced by many factors, such as the fracture pattern, tuberosity involvement, bone quality, surgeon preference, and the age and activity level of the patient [1].

HA is a well-known procedure for managing proximal humeral fractures. However, with the advent of RSA, the use of HA has become debatable owing to its unpredictable and non-homogeneous outcomes, its technical difficulties, and the fact that it is rarely indicated [2]. Moreover, HA is associated with a high rate of tuberosity complications (up to 50%) [2].

Failure of HA has been attributed to several causes including pain, deep infection, impaired shoulder function, rotator cuff degeneration, cartilage wear of the glenoid, aseptic loosening of the humeral component, and implant instability or dislocation [3,4]. Failed HA may be revised in several ways. Typical treatment options include revision to conventional total shoulder arthroplasty, revision to RSA, and resection arthroplasty [5,6,7,8,9]. Revision to RSA may be carried out by exchanging the complete hemiprosthesis (traditional revision RSA) or by retaining the existing humeral stem (conversion RSA). Indicated primarily for concomitant rotator cuff dysfunction, revision to RSA improves shoulder function and relieves pain; however, it is associated with a high complication rate [6,7,9,10,11].

The removal of the humeral stem is associated with a relatively high risk of humeral shaft fracture, which may jeopardize the anchoring of the new stem and result in implant failure [11,12,13]. Furthermore, removal of a well-fixed stem can be technically difficult, especially in cases where there is poor bone quality, and may require osteotomy of the humeral shaft [10,11,14]. Levy et al. observed that the high rate (32%) of prosthesis-related complications in their study of traditional revision RSA with stem exchange for failed HA was mainly associated with bone loss in the proximal humerus, glenoid, or both [7]. Therefore, to prevent bone loss and reduce the risk of intra- and postoperative complications, it would be best to retain the hemiprosthesis stem [10,14,15].

With the advent of modular convertible shoulder prosthesis systems, retaining the humeral stem during revision of HA to RSA has become a viable option. Indeed, the authors of some studies have shown encouraging results accompanied by reduced rates of complications and implant failures [13,14,15,16,17,18]. Moreover, clinical data have suggested that minimal changes in height and offset [14], shorter operating times [11,13,15], fewer intraoperative complications [11,13,15], and fewer subsequent revision surgeries [10,11,15] are associated with conversion RSA with modular prostheses compared to traditional revision RSA with stem exchange. However, these data are limited to a few short- and mid-term studies, primarily because revision of HA to RSA is a rare indication.

Our aim with this study is to provide further insight into the outcomes of this rare procedure by reporting the mid-term clinical results of an ongoing case series on conversion RSA with a modular prosthesis system in patients revised for failed HA after a proximal humeral fracture.

## 2. Materials and Methods

### 2.1. Study Design and Patient Selection

This study was a subgroup analysis of an ongoing, multicenter, prospective cohort study of 519 patients with various indications and operated with an Affinis Inverse or Affinis Facture Inverse system (Mathys Ltd., Bettlach, Switzerland) between 6 May 2008 and 1 June 2015. Of these, 17 patients met the following inclusion criteria: revision required for failed HA after proximal humeral fracture, Affinis Fracture or Articula system (Mathys Ltd., Bettlach, Switzerland) implanted as primary prosthesis, and stem removal not required during revision. Reasons for revision included secondary tuberosity dislocation or malunion after primary arthroplasty, luxation or instability after primary arthroplasty, rotator cuff disorder after primary arthroplasty, and glenoid erosion after HA, all resulting in pain and/or loss of function.

Conversion RSA was considered in patients who showed no stem loosening or infection and had adequate soft tissue tension for humeral stem retention, as assessed intraoperatively. Thirteen patients, enrolled from four centers in Germany and Switzerland, were considered suitable for conversion RSA (Figure 1). Of these patients, two were lost due to early revision, two to dementia or old age, and one to dislocation, leaving eight patients at the final follow-up. In the remaining four patients, conversion was not possible due to overstuffing; these patients were treated with traditional revision RSA with stem exchange.

Patients underwent a clinical examination preoperatively, after one year, and at the final follow-up. The mean final follow-up period was 55.1 months (range: 12.0 to 91.1 months).

### 2.2. Implant Characterstics

HA was performed using a fracture prosthesis (Affinis Fracture or Articula system), which is part of a modular implant system that allows conversion to RSA using the same humeral stem as in HA. For conversion RSA, the head component and the metaphyseal element were replaced with a reverse metaphyseal element (Affinis Fracture Inverse system) (Figure 2). This element can be fixed to the humeral stem with 10 mm of possible height adjustment and free torsion. On the glenoid side, either a 39 or 42 mm glenosphere can be chosen. The reverse metaphyseal element exists in a 0 or 3 mm offset version. If both metaphyseal parts are at the same level, converting HA to RSA would lengthen the humerus by 23 mm and medialize the center of rotation by at least 19 mm (from 4 to 23 or 25 mm lateral offset, Figure 3). This equates to the minimal lengthening in a worst-case scenario, where the central part was placed completely distally during the implantation of the fracture HA and revised with a 39 mm glenosphere and a 0 mm offset reverse metaphyseal element. When placing the reverse metaphyseal element completely proximally during the revision surgery with the largest glenosphere and the highest offset, a maximum lengthening of 38 mm and medialization of the center of rotation of 21 mm can be achieved.

### 2.3. Operating Technique

Surgeons used a standard deltopectoral approach in all cases. The recording of the operating time began at the moment of the first incision and ended at postoperative skin closure.

Initially, the humeral head was mobilized, and the metaphyseal part of the prosthesis was freed from the tuberosities if present. Next, the prosthetic head and the metaphyseal part were removed. On the humeral side, we removed the cement and bone from the proximal end of the stem enough to allow the insertion of the trial prosthetic epiphysis at the deepest position possible with the prosthetic system.

On the opposite side of the joint, we exposed the glenoid, then reamed and drilled it so that the metaglene could be inserted using a standard technique. Trial glenospheres and epiphyses were used to confirm the optimal configuration of the prosthesis before the definitive prosthetic components were implanted. Bony parts of the greater and lesser tuberosities, if present, were then reattached to the prosthetic epiphysis with heavy, non-absorbable sutures.

### 2.4. Clinical Evaluation

We measured the Constant score, American Shoulder and Elbow Surgeons (ASES) Shoulder Score, visual analogue scale (VAS) for pain and satisfaction, and range of motion preoperatively, after one year, and at the final follow-up. All complications were carefully monitored and recorded throughout the study.

### 2.5. Statistical Analysis

We compared the pre- and postoperative clinical parameters using a two-sided exact Wilcoxon rank sum test. In all cases, *p* values < 0.05 were considered significant. We presented the data as the mean (range) unless otherwise indicated. In cases of loss to follow-up (including death), we used the last observation for calculation.

## 3. Results

The patient demographics and characteristics are detailed in Table 1. The mean age of patients at the time of surgery was 73.6 years (range: 64.9 to 89.6 years), the mean final follow-up period was 55.1 months (range: 12.0 to 91.1 months), and the mean duration of surgery was 118.4 (range: 80.0 to 140.0) minutes.

### 3.1. Clinical Outcomes

The clinical outcomes, such as the Constant score, ASES Shoulder Score, VAS for pain and satisfaction, and ROM, improved significantly, both at one year postoperatively (*p* < 0.05) and at the last follow-up visit (*p* < 0.05) (Table 2).

### 3.2. Complications

Three patients presented with complications. One patient had a late infection at 5.2 months postoperatively and was treated by debridement and exchange of the bearing components; the patient had no clinical signs of ongoing infection at the last follow-up. Two patients had aseptic stem loosening. One occurred 49.2 months postoperatively and was treated by stem exchange. The other occurred 7.5 months postoperatively, and the humeral component was revised. However, the implant was later removed, and the patient was treated with antibiotics due to a late infection occurring 26.0 months after stem revision.

## 4. Discussion

In this study, we evaluated the conversion of failed HA to RSA using a modular prosthetic system and assessed the clinical outcomes at different time intervals postoperatively. We reported significant postoperative improvements in the Constant score, ASES Shoulder Score, VAS for pain and satisfaction, and ROM.

During the revision of failed HA, the correct balancing of soft tissue tension is important. Insufficient tension can result in instability of the prosthesis, whereas excessive tension can result in acromial fracture, poor deltoid function, and/or neurological lesions [19]. Moreover, restoring adequate tension can be complicated by scarring and altered muscle function. However, most current modular prostheses allow the height of the humeral head to be adjusted enough to modify the compressive forces across the glenohumeral joint, and thereby achieve adequate soft tissue tension.

Postoperative arm lengthening is a typical occurrence after RSA. Lädermann et al. reported a mean postoperative lengthening of 23 ± 12 mm in primary and revision RSA using a different prosthesis than we did [19], while Teschner et al. measured a mean increase in implant height of 24 ± 2.6 mm in the conversion of failed HA to RSA using the same prosthesis as we did [14]. Both of these values are similar to the lengthening that can be achieved with the reverse modular prosthesis used in our series (23 mm) (Figure 3). Moreover, we did not observe any complications related to the under- or over-tensioning of the deltoid tissue. We therefore believe that the modular prosthesis used in our series allowed for the optimal soft tissue balance required for retaining a firmly implanted humeral stem.

Although several modular convertible shoulder prostheses are available today for the successful conversion of HA to RSA [14], stem retention may not always be possible, especially when the humeral stem is poorly positioned or not well fixed [10,11]. In these cases, stem revision is needed [10,11]. Similarly, conversion RSA is not feasible in cases with infection or where the stem cannot be distalized enough, as seen in the present study. Finally, the use of stemless anatomical implants in primary procedures does not allow conversion RSA. We also found that not every patient in the present study could benefit from conversion RSA. Of the 17 cases where conversion RSA was considered, four cases (23%) could not be converted due to overstuffing. A similar rate was found in a recent study where 22% of the patients could not be converted despite the presence of a modular prosthesis [10], indicating a limitation of today’s convertible modular prosthetic systems. Other limitations include the need for highly skilled surgeons and a meticulous surgical technique for implanting a convertible modular prosthetic system during the primary procedure [16]. These limitations necessitate the continued use of modular non-convertible shoulder prostheses.

Revision surgery of failed HA remains technically challenging and may be associated with less predictable clinical outcomes and various complications [18]. Nevertheless, in the current study, we found a significant improvement in the mean Constant score (from 21.7 preoperatively to 57.9 at the last follow-up; *p* = 0.0001). Other studies on the revision of failed HA to RSA also found similar increases in the mean Constant scores: from 12.67 to 45.08 [6] and from 8.9 to 41.0 [18].

Besides the functional scores, intraoperative parameters such as the operating time are also important contributors to successful conversion RSA. In recent studies comparing conversions and revisions, conversion procedures resulted in shorter operating times [10,11,13,15]. We found similar results in our study: the mean operating time for conversion RSA was shorter than that for traditional revision RSA [118.4 min (range: 80.0 to 140.0) versus 150.0 min (range: 100.0 to 230.0), our unpublished results].

Despite successful clinical outcomes, complication rates remain high after conversion RSA, ranging from 22% to 43% [11]. We found complications in three (23%) patients who underwent conversion RSA. Revision with stem exchange was needed in two (15%) patients due to aseptic loosening. Although we observed an overall complication rate similar to that reported in other studies of RSA after failed shoulder arthroplasty [6,20,21], more clinical evidence is needed to draw definite conclusions in this regard. Previous studies also showed that modular prosthesis designs minimized the risk of periprosthetic humeral shaft fractures [18]. We confirmed this finding; no periprosthetic fractures or other implant-related complications were observed in our study.

Our study had a number of limitations; these included the absence of standardized radiographs for the accurate measuring of arm length and the small sample size owing to the nature of the indication, which limited the interpretation of our findings. This is why—although challenging—studies with larger patient cohorts are needed to investigate the full potential of modular systems in the conversion of failed HA to RSA. Finally, the results from the type of modular convertible shoulder prosthesis used in our study are not sufficient to make generalized conclusions on the outcomes of conversion RSA. For this, a clinical comparison with different implant types will be needed.

## 5. Conclusions

Modular shoulder arthroplasty is a suitable procedure for the conversion of failed HA to RSA in elderly patients, allowing a successful conversion in the majority of cases. In our study, all measured postoperative clinical outcomes improved significantly. Moreover, retaining the prosthetic stem was associated with a short operating time. Finally, the observed complication rate was acceptable, and no prosthesis-related complications occurred. Although not feasible in every case, conversion RSA is a viable treatment option in the elderly, which can provide successful mid-term results with an acceptable complication rate. Nevertheless, studies with larger sample sizes should be carried out to provide greater insight into conversion RSA outcomes in this rare indication.

## Figures and Tables

**Figure 1 jcm-11-00834-f001:**
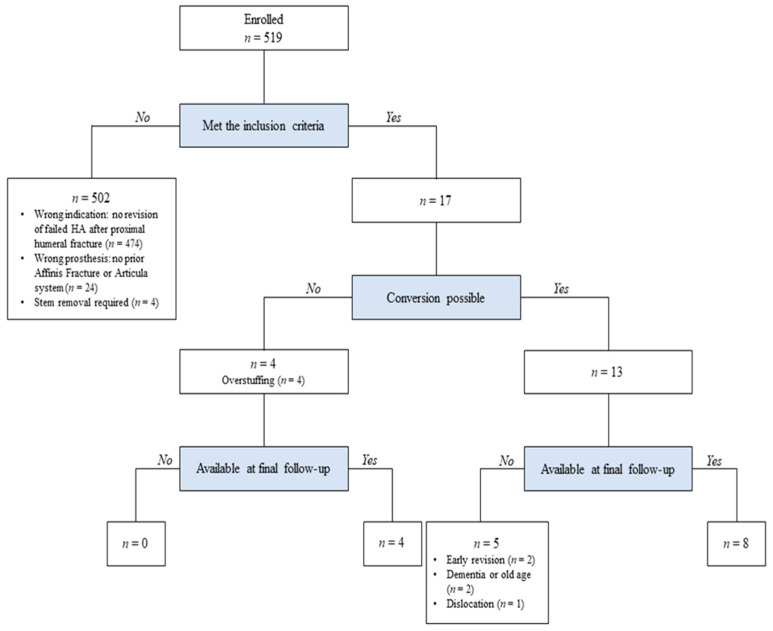
Flow diagram of the patient selection.

**Figure 2 jcm-11-00834-f002:**
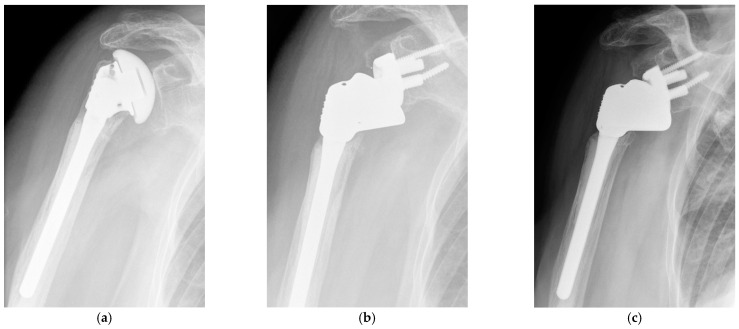
Failure of hemiarthroplasty and conversion to reverse shoulder arthroplasty. (**a**) Preoperative anteroposterior radiograph of a hemiarthroplasty with a firmly cemented stem showing insufficiency of the supraspinatus tendon and superior migration of the humeral head; (**b**) Postoperative anteroposterior radiograph of a conversion reverse shoulder arthroplasty with humeral stem retention; (**c**) Four-year postoperative anteroposterior radiograph showing prosthesis in situ.

**Figure 3 jcm-11-00834-f003:**
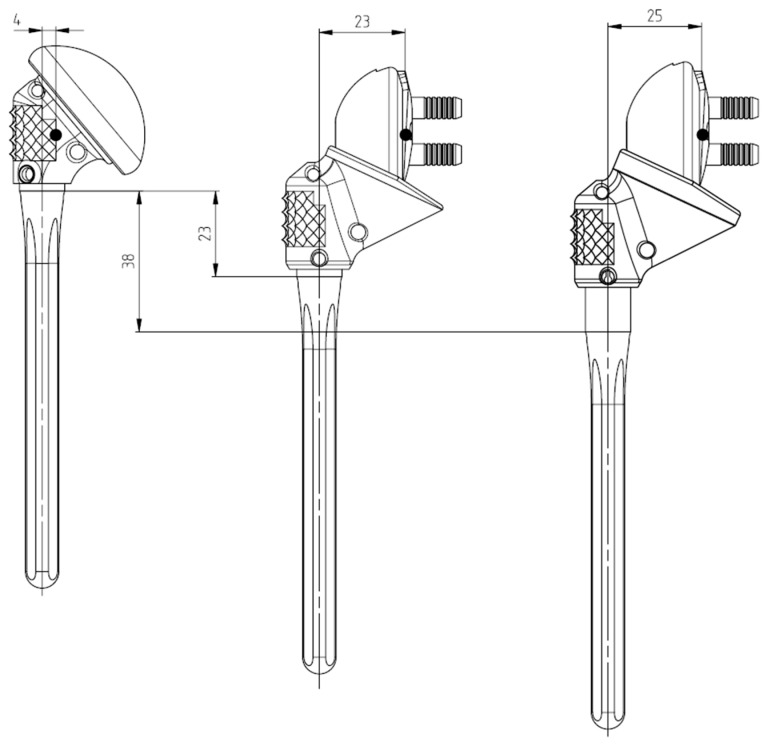
Modular reverse shoulder prosthesis. If both metaphyseal parts are at the same level, conversion leads to 23 mm of distalization and medializes the center of rotation by 19 mm (left). This corresponds to the worst-case scenario if the reverse metaphyseal element was placed most distally during the primary implantation of hemiarthroplasty (middle). The maximum lengthening of the conversion is 38 mm by choosing a 42 mm glenosphere and an offset of 3 mm (right). The length of the anatomic and the reverse prosthesis can be adjusted by 10 mm to reduce distalization. The center of rotation is medialized between 19 mm and 21 mm.

**Table 1 jcm-11-00834-t001:** Patient demographics and characteristics.

Variable	Value
Sex (female/male)	9/4
Operated side (right/left)	6/7
Age at surgery (years)	73.6 (64.9–89.6)
Time since implantation of hemiprosthesis (months)	16.7 (3.9–61.7)
Follow-up period (months)	55.1 (12.0–91.1)

The values of the age at surgery, time since implantation of hemiprosthesis, and follow-up period are reported as means (ranges). Total number of patients: 13.

**Table 2 jcm-11-00834-t002:** Clinical outcomes.

Clinical Outcome	Preoperative (*n* = 13)	At 12 Months (*n* = 11)	*p* Value *	At Last Follow-up (*n* = 8)	*p* Value **
Constant score	21.7 (4.0–52.0)	50.1 (37.0–71.0)	0.0001	57.9 (42.0–96.0)	0.0001
ASES Shoulder Score	24.3 (6.7–46.8)	63.8 (46.7–86.7)	<0.001	66.9 (46.7–98.3)	<0.001
VAS for pain	7.3 (5.0–9.0)	2.3 (0.0–5.0)	<0.001	2.3 (0.0–5.0)	<0.001
VAS for satisfaction	2.0 (0.0–6.2)	7.9 (5.0–10.0)	<0.001	8.0 (7.0–10.0)	<0.001
Active ROM in abduction (°)	38.8 (10.0–100.0)	103.2 (50.0–180.0)	<0.001	111.9 (50.0–160.0)	0.006
Active ROM in forward flexion (°)	46.2 (10.0–130.0)	111.8 (60.0–160.0)	0.0001	122.5 (60.0–180.0)	0.001

Values reported as means (ranges). *p* values from two-sided exact Wilcoxon rank sum test between preoperative and 12-month follow-up (*) and between preoperative and last follow-up (**). ASES: American Shoulder and Elbow Surgeons; n: number of patients; ROM: range of motion; VAS: visual analogue scale.

## Data Availability

The data presented in this study are available on request from the corresponding author. The data are not publicly available due to ethical restrictions.

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
