# Peer review of "Conversion of Hemiarthroplasty to Reverse Shoulder Arthroplasty with Humeral Stem Retention"

_jcm, 2022, doi:10.3390/jcm11030834_

Round 1

Reviewer 1 Report

In the paper, the authors mainly investigated the efficacy of conversion reverse shoulder arthroplasty (RSA) for the patients with failed hemiarthroplasty after proximal humeral fracture. They found that RSA could be a promising alternative option for the elderly patients. The paper is well-organized, and written in a logical pattern. The paper needs minor revision, such as text typo error and polish.

Author Response

In the paper, the authors mainly investigated the efficacy of conversion reverse shoulder arthroplasty (RSA) for the patients with failed hemiarthroplasty after proximal humeral fracture. They found that RSA could be a promising alternative option for the elderly patients. The paper is well-organized, and written in a logical pattern. The paper needs minor revision, such as text typo error and polish.

Answer: Thank you for this comment. The manuscript has been revised thoroughly to address all reviewers’ comments.

Reviewer 2 Report

I thank the authors and the editor to give me the opportunity to revise this manuscript. The authors aimed to evaluate the mid-term clinical results of 17 elderly patients revised for failed hemiarthroplasty. Of them, 13 underwent conversion in reverse shoulder arthroplasty with a modular prosthesis system, preserving the HA stem. Among these 13 patients, 2 were lost due to early revision, 2 to dementia or old age, and 1 to dislocation, leaving 8 patients at final follow-up. Among these 8 patients 3 had complications, 1 infection and 2 aseptic loosening.

So far, 5 patients out of 17 with failed HA had successful outcomes after conversion to RSA. In contrast with the conclusion supported by the authors conversion in RSA of a failed HA seems an indication that should be proposed with extreme caution, due to the high risk of unfeasibility and failure. The authors should better clarify what the real failure rate of the technique is (apparently 6 total cases of failure). Moreover, the findings from a unique type of implant cannot be generalized to every type of implant.

In the discussion, the authors should remark the advantages, disadvantages and limits of RSA, and what are the exact criteria for defining HA failure in presence of a stable stem (pain, stiffness, instability, impaired function…) and for opting for RSA. Is the decision taken pre-operatively or intra-operatively?

I thank the authors for answering this questions and for commenting these issues.

Author Response

I thank the authors and the editor to give me the opportunity to revise this manuscript. The authors aimed to evaluate the mid-term clinical results of 17 elderly patients revised for failed hemiarthroplasty. Of them, 13 underwent conversion in reverse shoulder arthroplasty with a modular prosthesis system, preserving the HA stem. Among these 13 patients, 2 were lost due to early revision, 2 to dementia or old age, and 1 to dislocation, leaving 8 patients at final follow-up. Among these 8 patients 3 had complications, 1 infection and 2 aseptic loosening.

So far, 5 patients out of 17 with failed HA had successful outcomes after conversion to RSA. In contrast with the conclusion supported by the authors conversion in RSA of a failed HA seems an indication that should be proposed with extreme caution, due to the high risk of unfeasibility and failure. The authors should better clarify what the real failure rate of the technique is (apparently 6 total cases of failure). Moreover, the findings from a unique type of implant cannot be generalized to every type of implant.

Answer: Thank you for this comment. Based on our results, failure (defined as need for revision) was seen in two (15%) cases where stem revision was necessary due to aseptic loosening, which is outlined in the results section (lines 170 to 174). Nevertheless, to address your comment, we have elaborated on this point in the discussion section (lines 226 to 229).

We completely agree that each implant type is unique with regard to its convertibility options including lengthening and lateralization. The advantage of the implant used in this study is that surgeons can freely adjust the length and retroversion of the central part. This feature is important for converting an anatomical hemiprosthesis (retroversion of about 30°) to a reverse shoulder prosthesis (retroversion of 0° to 10°). To address your comment, “the findings from a unique type of implant cannot be generalized to every type of implant”, we have added this as limitation in the discussion section (lines 237 to 240).

In the discussion, the authors should remark the advantages, disadvantages and limits of RSA, and what are the exact criteria for defining HA failure in presence of a stable stem (pain, stiffness, instability, impaired function…) and for opting for RSA. Is the decision taken pre-operatively or intra-operatively?

I thank the authors for answering this questions and for commenting these issues.

Answer: Thank you for raising this point. The advantages of conversion RSA are already part of the introduction (lines 59 to 66) and discussion sections (lines 218 to 223). For the disadvantages and limitations of conversion RSA, we have expanded the discussion accordingly (lines 197 to 203, 208 to 211). In addition, we have added the reasons for revision of HA to the methods section (lines 80 to 83). The criteria for choosing conversion RSA are presented in the methods section (lines 84 to 85). Finally, the decision to carry out conversion RSA was made intraoperatively (lines 84 to 86).

Reviewer 3 Report

The manuscript is well written and structured, the authors have adequate background and the topic is of great interest to a very small niche of specialty orthopedics - shoulder surgeons.

 My discretionary comments are as follows:

The authors present 13 cases of stem retention and 4 of stem exchange, as 'case series' - subgroup analysis of a larger prospective study. They than use the Wilcoxon rank sum test to compare the outcome of the patients preoperatively, after surgery and at the last followup. Can one really make a statistical analysis with any common test comparing the change in status of as little as 8 to 13 patients? Is the statistical analysis of any relevance? To address a change in the status of the same patients why didn't they use ANOVA? I am not proficient in statistics but I think for such a low number of subjects one can only use descriptive statistics. They have 2 groups of 13 and 4 to compare which are also too small. To present that the preoperative state is improved after surgery is to be expected. To what level of improvement compared to other alternative scenarios such as primary HA, primary RSA, other revision RSA would be more relevant.

Given the low number of patients, the manuscript's main value is to include complete datasets for future use in systematic reviews. 

The discussion can be expanded. The authors may add more about why is stem retention not always possible, what are the limitations of current systems, what are the most commonly used systems which allow stem retention, why do we still use systems that cannot be converted to a reverse with stem retention.

Sincerely,

Author Response

The manuscript is well written and structured, the authors have adequate background and the topic is of great interest to a very small niche of specialty orthopedics - shoulder surgeons.

 My discretionary comments are as follows:

The authors present 13 cases of stem retention and 4 of stem exchange, as 'case series' - subgroup analysis of a larger prospective study. They than use the Wilcoxon rank sum test to compare the outcome of the patients preoperatively, after surgery and at the last followup. Can one really make a statistical analysis with any common test comparing the change in status of as little as 8 to 13 patients? Is the statistical analysis of any relevance? To address a change in the status of the same patients why didn't they use ANOVA? I am not proficient in statistics but I think for such a low number of subjects one can only use descriptive statistics. They have 2 groups of 13 and 4 to compare which are also too small. To present that the preoperative state is improved after surgery is to be expected. To what level of improvement compared to other alternative scenarios such as primary HA, primary RSA, other revision RSA would be more relevant.

Answer: Thank you for this comment. We completely agree that the number of cases is too low for meaningful statistical conclusions. For this reason, no between-group comparisons have been made. However, we used the Wilcoxon rank sum test for within-group comparisons of clinical outcomes at each follow-up time point for patients in the conversion RSA group mainly because clinical improvement may not always be evident for patients undergoing revision. ANOVA is used to compare means of two or more groups and requires normal data distribution, which is not the case in our study due to the small sample size. Therefore, we considered the Wilcoxon rank sum test to be appropriate.

While our study did not directly compare conversion RSA to revision RSA in failed HA, we found similar improvements in mean Constant scores as reported after revision of failed HA to RSA (lines 215 to 217).

Given the low number of patients, the manuscript's main value is to include complete datasets for future use in systematic reviews.

Answer: Thank you for raising this point. We have added a table (Table A) below for your review. While looking at the data again in more detail, we realized that adding the complete dataset may only be of limited value for future systematic reviews because of the small sample size. As established above, statistically significant conclusions between the two groups cannot be made. Nevertheless, we are happy to include the table below in the revised manuscript if needed.

Stem retained

Gender

Age (years)

Time to revision (months)

CS

ASES Shoulder Score

Forward flexion (°)

Abduction (°)

VAS for pain

VAS for satisfaction

Reason for revision of HA

Yes

m

74

5

4

6.67

15

15

9

0

Rotator cuff disorder after primary arthroplasty and joint instability

Yes

f

66

11

16

16.67

70

40

7

1

Joint instability

Yes

m

70

10

18

15

20

10

9

2

Glenoid erosion after HA

Yes

f

65

22

32

40

80

70

6

2

Rotator cuff disorder after primary arthroplasty

Yes

m

68

10

17

10

35

30

8

1

Secondary tuberosity dislocation

Yes

f

83

28

16

30

30

20

5

1

Joint instability

Yes

f

78

17

36

33.33

70

60

6

4

Rotator cuff disorder after primary arthroplasty

Yes

f

90

18

8

11.67

40

30

9

0

Secondary tuberosity dislocation

Yes

f

70

4

20

25

30

30

7

0

Secondary tuberosity dislocation

Yes

f

76

17

13

15

10

30

8

3

Secondary tuberosity dislocation and joint instability

Yes

f

75

62

52

39.17

130

100

7.5

6.2

Rotator cuff disorder after primary arthroplasty

Yes

m

69

8

37

46.83

50

40

7.3

2.7

Joint instability

Yes

f

72

7

13

26.17

20

30

6.1

3.1

Rotator cuff disorder after primary arthroplasty and joint instability

No

m

72

12

16

21.67

30

30

6

0

Secondary tuberosity dislocation

No

m

72

12

4

8.33

30

30

9

0

Secondary tuberosity dislocation

No

f

85

66

17

31.67

40

20

5

2

Rotator cuff disorder after primary arthroplasty, secondary tuberosity dislocation and joint instability

No

f

67

5

4

10

30

30

9

1

Secondary tuberosity dislocation

Table A. Complete dataset of patients undergoing revision with or without stem retention. CS: Constant score, ASES: American Shoulder and Elbow Surgeons, VAS: visual analogue scale, HA: hemiarthroplasty, m: male, f: female

The discussion can be expanded. The authors may add more about why is stem retention not always possible, what are the limitations of current systems, what are the most commonly used systems which allow stem retention, why do we still use systems that cannot be converted to a reverse with stem retention.

Answer: Thank you for this comment. We have expanded the discussion including the reasons of why stem retention may not always be possible, the limitations of current modular convertible systems, and the need of using modular non-convertible systems (lines 197 to 203, 208 to 211). Regarding the discussion of the most used modular convertible systems today, we believe that discussing the technical details of contemporary modular convertible systems would be exhaustive and beyond the scope of the present study, which is why we have not elaborated on this point in more detail.